# The Effects of Ultrasound-Guided Intra-Articular Injections with Hyaluronic Acid and Corticosteroids in Patients with Hip Osteoarthritis: A Long-Term Real-World Analysis

**DOI:** 10.3390/jcm12206600

**Published:** 2023-10-18

**Authors:** Gianpaolo Ronconi, Sefora Codazza, Maurizio Panunzio, Fabiana La Cagnina, Mariantonietta Ariani, Dario Mattia Gatto, Daniele Coraci, Paola Emilia Ferrara

**Affiliations:** 1Department of Rehabilitation, Catholic University of Sacred Heart, 00168 Rome, Italy; gianpaolo.ronconi@policlinicogemelli.it; 2Department of Ageing, Neurosciences, Head-Neck and Orthopaedics Sciences, University Polyclinic Foundation Agostino Gemelli IRCSS, 00168 Rome, Italy; mariantonietta.ariani01@icatt.it (M.A.); dariomattia.gatto01@icatt.it (D.M.G.); paolaemilia.ferrara@policlinicogemelli.it (P.E.F.); 3Responsible Research Hospital, 86100 Campobasso, Italy; m.panunzio@responsiblecapital.ch; 4Physical and Rehabilitation Medicine, University of Rome Tor Vergata, 00133 Rome, Italy; fabiana.lacagnina@students.uniroma2.eu; 5Department of Neuroscience, Section of Rehabilitation, University of Padova, 35122 Padova, Italy; daniele.coraci@unipd.it

**Keywords:** hip osteoarthritis, ultrasound-guided intra-articular injections, hyaluronic acid, corticosteroids

## Abstract

Intra-articular (IA) ultrasound-guided hip injections are currently considered a cornerstone of the conservative management of symptomatic hip osteoarthritis (HOA), although their effect on clinical outcomes has not been fully elucidated.The purpose of this study is to investigate the effectiveness of ultrasound-guided IA hip injections of hyaluronic acid (HA) with or without corticosteroids (CS) on pain relief and functional improvement in patients with HOA. In total, 167 patients with HOA were assessed at baseline (T0) and 12 months after injection (T1) using the VAS and GLFS scores. The sample consisted mainly of female subjects (58.1%), presenting an average age of 70.6 ± 12.2 years and grade 3 HOA (63.9%) according to the Kellgren–Lawrence classification. Most of the patients (76.2%) underwent unilateral hip injection with a combination of medium-high molecular weight HA (1500–2000 kDa) and CS. At T1, lower use of anti-inflammatory drugs, an increase in the consumption of chondroprotectors, and an overall reduction of instrumental physical therapies and therapeutic exercise were recorded. In addition, a statistically significant intragroup and between-group decrease observed at T1 for both the VAS and GLFS. Study results suggested that intra-articular hip injections with HA alone and with CS could represent a useful therapeutic tool for pain reduction and functional improvement for patients with hip osteoarthritis.

## 1. Introduction

Osteoarthritis (OA) is the most common joint disease, affecting about 400 million people worldwide [1]. The improvement in life expectancy and the increase in the average age of the general population causes its prevalence to show a marked increase [2].

The significant social impact of OA seriously and progressively invalidates a patient’s quality of life and determines a substantial rise in public health costs as a consequence of impaired work productivity and early retirement [2], representing one of the major causes of chronic pain and disability in older adults [3].

Osteoarthritis typically affects the weight-bearing joints, with the hip being the second most frequently involved large joint after the knee [4].

Hip osteoarthritis (HOA) is one of the most frequent osteoarticular conditions affecting the elderly population [5]. Radiographic evidence of HOA is found in nearly 5% of subjects aged over 65 years [6], and the risk of developing symptomatic hip OA is approximately 25% [7]. Hip osteoarthritis (HOA) has one of the highest financial burdens [8], and the prevalence ranges from 6.7% to 9.2% among adults 45 years of age [9] and increases to 25% in patients aged over 55 years, constituting a source of chronic joint pain and stiffness [10]. OA affecting the hip joint shows a particular etiologycompared to other joint sites involved [5], linked to the femoro-acetabular morphology [11,12], which can determine the development of abnormal shear forces responsible for the onset of the chronic inflammatory process [13,14].

Nowadays, the prevalence of total hip arthroplasty is increasing [15]. This means that resolutive non-surgical therapeutic options are still limited and often controversial.

The conservative management of HOA is based on a combination of nonpharmacological and pharmacological modalities [16,17]. Among those, intra-articular (IA) hip injections are currently considered a cornerstone of the conservative management of symptomatic hip osteoarthritis, although their effect on clinical outcomes has not been fully elucidated.

The IA administration of drugs in the hip joint is increasingly considered a safe and effective therapeutic tool to reduce pain and improve function in hip osteoarthritis [18], with a reported complication rate of between 10 and 30% of patients [19], which is significantly decreased by the introduction of ultrasound guidance [20].

Intra-articular injections have become commonly recommended in the management of large joint osteoarthritis [16,17]. The infiltrative treatment has been used widely in hip OA despite the paucity of available data, compared with knee OA, for which the subject has been extensively studied. In fact, only a few trials of poor quality have been conducted in symptomatic hip OA during the last two decades, and so only scarce evidence is currently presented in the literature [18].

Different medications can be injected into the hip joint with different levels of evidence-based efficacy, depending on whether they intervene more in the control of inflammation, pain, or mechanical function [18]. The two main substances used in recent times for intra-articular hip injections are corticosteroids (CS) and hyaluronic acid (HA).

Corticosteroids have been shown to be effective in decreasing pain in symptomatic hip osteoarthritis [21]. Intra-articular corticosteroid injections are commonly used in clinical practice, especially in the case of contraindications or non-response to oral non-steroidal anti-inflammatory drugs (NSAIDs), relieving pain typically for a few weeks [22]. However, the number of randomized clinical trials is limited, and the quality of the evidence remains relatively poor. While IA CS are recommended for knee OA [23], such a recommendation has not been indicated for the hip where non-pharmacological treatments are generally preferred over pharmacological interventions [24].

Moreover, the latest available guidelines issued by the OARSI in 2019 provided no recommendation for IA steroid administration in the hip, showing a certain distrust towards pharmacologic interventions in hip OA [25].

The infiltration strategy with CS is often distanced because of important risk factors, such as local infection and soft tissue irritation at the site of injection, the exacerbation of pain, septic arthritis, and chondrotoxicity [26].

Hyaluronic acid is the most used substance in intra-articular infiltrations of the hip or knee and guarantees visco-integration, which seems to have both mechanical and biological effects by restoring the visco-elasticity of the synovial fluid and the lubrication of the joints [27]. Another effect of hyaluronic acid is represented by the down-regulation of pro-inflammatory cytokines, such as interleukin-1 (IL-1), impacting pain relief and immune modulation. According to some studies conducted in vitro and in vivo on humans, hyaluronic acid also appears to have a chondroprotective effect [28].

Specifically, with regard to HA, in the last two decades, viscosupplementation has become popular in clinical practice because of its notable ability to reduce pain, improve function, and possibly delay definitive arthroplasty [29]. Although the positive clinical effects of HA have been demonstrated in the knee joint, the latest literature recommends otherwise for its use in the hip because the highest quality trials showed weaker evidence regarding the efficacy of HA injections, and the guidelines of European and international scientific societies for the management of hip osteoarthritis differ regarding recommendations of intra-articular HA in terms of the types and injective protocols [25,30,31,32,33,34,35].

At best, it has shown a promising effect in moderate-grade hip OA, as shown by Pogliacomi et al. [36], who reported the efficacy of intra-articular HI of a single dose of high-weight HA in patients under a follow-up of 12 months.

In the literature, there are currently no studies about the intra-articular administration of the association of HA and CS in the hip joint, with the exception of the study by De Rezende et al. [37], which showed that the combination of the two intra-articular drugs provided better improvement for pain, function, and quality of life compared to HA alone until the 1-year follow-up. Thus, it appears important to assess the real efficacy and tolerability of IA injections in hip OA with the association of CS and HA in order to obtain a conservative therapeutic weapon for the management of symptomatic hip osteoarthritis in the short and long term.

The aim of this retrospective observational real-life study is to investigate the long-term effectiveness of ultrasound-guided intra-articular hip injections of hyaluronic acid, alone and in association with corticosteroids, for pain relief and functional improvement in patients with hip osteoarthritis.

## 2. Materials and Methods

The study was conducted in accordance with the ethical standards of the 1964 Declaration of Helsinki and its later amendments or comparable ethical standards (Protocol number 0039224/21 of 8 November 2021).

This retrospective observational study analyzes the data obtained from the medical records of patients with moderate hip osteoarthritis treated with ultrasound-guided hip injections with hyaluronic acid and corticosteroids, alone or in combination, from January 2020 to December 2022, and access was obtained to the outpatient clinics of the Inpatient and Rehabilitation Services Department of the University Polyclinic Foundation Agostino Gemelli of Rome.

Informed consent was obtained from all the subjects involved in the study. The participants could withdraw their consent to participate at any time without any consequences.

The inclusion criteria were as follows: OA grade > 2 on the Kellgren–Lawrence scale [38], Mini Mental State Examination (MMSE) score > 23 [39], patients who signed the informed consent to injection and privacy.

The exclusion criteria were as follows: age < 18 years, patients with hip prostheses on the affected limb, MMSE < 23, signs of joint infection, severe inflammation, skin disorders or infection at the injection site.

The patients were evaluated at baseline before the infiltration treatment (T0) and after 12 months (T1) by telephone interview. Personal and clinical information (age, gender OA diagnosis date, traumatic events, comorbidities, current therapies), pharmacological therapies in progress, ongoing anti-inflammatory and pain-relieving therapies, and rehabilitative treatments in progress related to hip osteoarthritis were collected.

Pain assessment was performed using the VAS scale, and the degree of disability and the level of autonomy were measured using the Geriatric Locomotive Function Scale (GLFS-5) [40] at baseline (T0) and after 12 months (T1) using a telephone questionnaire.

### Statistical Analysis

The statistical analysis was performed using a data processing computer system (SPSS version 14 statistical software developed by Norman H. Nie, Dale H. Bent, C. Hadlai Hull. License Trialware or SaaS). For the ordinal and nominal data, descriptive data are reported. For the continuous-interval data, the mean and standard deviation are reported. The comparison between the data was performed using the student’s *t* test for the independent variables and the X^2^ test for the categorical variables. The assumed confidence level was 95% and the significance was considered for *p* ≤ 0.05.

## 3. Results

Table 1 shows the characteristics of the sample, made up of 167 subjects, of which 41.9% were male and 58.1% were female, with an average age of 70.6 ± 12.2 years. Overall, the patients received monolateral intra-articular injections (87.2%); a total of 12.8% of patients underwent intra-articular bilateral injections.

With regard to the molecular characteristics of hyaluronic acid, a medium-high molecular weight (1500–2000 kDa) was used for most of the patients (96%), while 4% of the patients were treated with low molecular weight hyaluronic acid (620–1200 kDa).

In 76.2% of the patients, hyaluronic acid was injected in combination with a corticosteroid (methylprednisolone acetate 40 mg/L mL), while in 23.8% of the patients, only hyaluronic acid was used. The majority of patients (63.9%) had grade 3 hip osteoarthritis according to Kellgren–Lawrence radiological grading [38], while 0.7% of the patients had grade 1, 18.4% had grade 2, and 17% had grade 4. Regarding the pre-treatment outcome measures of pain and functional status, our data show a mean VAS of 6.30 ± 2.36 and a GLFS of 60 ± 13.11 at the time of the first injection (T0).

Table 2 shows the data referring to the baseline questionnaire (T0) and the subsequent re-administration 12 months later (T1) during the telephone interview, in which 88 out of 167 patients participated. Notably, there wasan increase in the percentage of patients who did not take any drugs, which varied from 42.5% at the baseline to 69.8% after 1 year. A lower use of drugs emerges from the baseline, with a decrease in paracetamol consumption from 19.4% at T0 to 10.5% at T1, a decrease in the consumption of NSAIDs from 33.1% at T0 to 18.6 at T1, and a decrease in the consumption of oral steroids from 5% at T0 to 1.2% at T1. Furthermore, an overall increase in the consumption of nutraceuticals was recorded at T1, with a decrease in the non-user subjects from 73.3% at T0 to 37.6% at T1 and an increase in the baseline consumers from 26.7% to 62.4% at T1.

With regard to rehabilitative intervention, the data showed a reduction in the patients who did not perform therapeutic exercise, from 87.3% at T0 to 37.5% at T1, and a parallel increase in the subjects who had already been prescribed a motor rehabilitation program, from 12.7% at T0 to 62.5% at T1. Also, an overall lower use of instrumental physical therapies was detected at T1, with a slight decrease in the percentage of patients who did not perform physical therapies, from 96% at T0 to 87% at T1, and an increase in the patients who performed them, from 4% at T0 to 12.3% at T1.

Finally, the percentage of patients who had not undergone hip arthroplasty was 99% at T0, with a reduction to 77.4% at T1. The percentage of patients who were operated on was zero at the baseline, with an increase of up to 20.2% after 12 months. The percentage of patients on the surgical waiting list went from 1% at T0 to 2.4% at T1.

The mean changes from the baseline (T0) over 12 months (T1) of the outcome measures are presented in Table 3. A statistically significant reduction of both the VAS and GLFS at T1 is noted: the VAS decreased from 6.3 at T0 to 4.81 at T1, and the GLFS ranged from 60 at T0 to 51.38 at T1.

Table 4 illustrates the trend in the clinical and functional outcome measures between the two groups of patients analyzed who were respectively infiltrated with hyaluronic acid alone or in association with corticosteroids. A statistically significant intragroup and between group reduction was observed for both the VAS and GLFS. For the patients infiltrated with the association of hyaluronic acid and cortisone, a statistically significant reduction in the VAS was observed, from 6.54 at T0 to 4.96 at T1, and also for the GLFS, which decreased from 56.67 at T0 to 48.61 at T1. The data from the subjects treated with hyaluronic acid alone showed a statistically significant reduction of the VAS, from 4.93 at T0 to 4.81 at T1, and the GLFS, from 56.67 at T0 to 48.61 at T1.

The two groups showed a different sample size, with 127 subjects for the HA + CS Group and 40 subjects for the HA Group at T0 and 79 subjects for the HA + CS Group and 9 subjects for the HA Group at T1.

## 4. Discussion

Nowadays, intra-articular injections have become commonly recommended in the management of large joint osteoarthritis, including the hip, which is generally injected under imaging guidance [20]. The demographic and clinical characteristics of our sample reflect the prevalent distribution of hip osteoarthritis taken into consideration in the most recent reviews on viscosupplementation and corticosteroid injections [24,25].

With regard to the demographic data, our sample consisted mostly of women with a mean age of 70.6 years, which is in line with the epidemiology currently known for hip osteoarthritis with a prevalence of 10–30% in people aged over 60 years [41], although it has been shown that age and gender were not significant predictors of response for IA CS [2].

Regarding the hip osteoarthritis phenotype possessed at baseline, it was a monolateral osteoarthritis, with moderate pain and mild functional impairment, referring to the VAS and GLFS scales, and of grade 3 according to Kellgren–Lawrence classification, with definite joint space narrowing, moderate osteophyte formation, some sclerosis, and possible deformity of the bone extremities. Radiographic severity has been related to the degree of clinical improvement provided by intra-articular CS injections in the hip [2,42]. Infact, it has been proven that the Kellgren–Lawrence grade is a good measure of treatment efficacy because patients with severe hip OA—that is, grades 3 and 4 based on the Kellgren–Lawrence scale—benefited the most from IA CS injections compared to patients with mild HOA [2].

So, looking at this possible relationship between radiographic damage and pain relief after IA corticosteroid injections in the hip OA, and based on the preliminary encouraging results showing pain and disability reduction with HA and CS association [37], we decided to associate CS with HA as a desirable standard infiltration protocol for painful Kellgren–Lawrence 3 grade hip osteoarthritis.

Various preparations of HA based on different molecular weights are commercially available, but there is no general consensus on which type is most effective. In our study, a medium-highmolecular weight HA was used, based on the consolidated evidence of the decrease in the concentration and size to 50–33% of HA in an arthritic joint [43] and following the international recommendations that high-molecular-weight HA might have better clinical efficacy in moderate–severe OA, reaching a high and durable concentration with low doses [44]. However, our previous review about intra-articular injections for hip OA [18] did not reveal a different improvement in pain and functional outcomes between patients treated with high-molecular-weight and mean-molecular-weight HA until 12 months after treatment.

In addition, regarding the use of high-molecular-weight HA (HMWHA), a recent review with metanalysis revealed that an intra-articular HMWHA injection provided pain relief, functional improvement, and no severe complications on an immediate short-term basis; however, the results do not favor treatment with HMWHA over other treatment methods [45].

From the literature review, the efficacy of hyaluronic acid and cortisone injected separately can be suggested, although evidence from RCTs comparing IA CS injections to viscosupplementation in the hip is quite scarce [24].

In light of the paucity of available data regarding the association of intra-articular HA and CS for the hip joint, our observational study has been primarily designed to evaluate the possible combination of the well-known short-term efficacy of CS with the supposed long-term efficacy of HA in order to design a possible injective treatment protocol for symptomatic HOA in terms of pain reduction and functional improvement.

The data on pain and function (Table 3) showed that in the context of conservative treatment, intra-articular hip injections with HA with or without CS have proved to be a useful tool in reducing some amount of pain in the longterm since the anti-inflammatory drug intake and manual and instrumental physiotherapy treatments decreased.

Indeed, from a retrospective analysis of the collected data, a statistically significant overall decrease in subjective pain and a functional improvement after one year (Table 3) can be observed, taking into account that most of the patients underwent intra-articular injection with HA and CS in combination, and only a smaller percentage of patients were treated with hyaluronic acid alone. In addition, an irrelevant percentage of the patients underwent prosthesis surgery (Table 2).

The analysis intragroup and between groups (Table 4) confirmed the efficacy of hyaluronic acid alone or in combination with corticosteroids at one year from injection in relieving pain and improving the motor skills of daily living.

No patients reported immediate or delayed adverse effects, confirming the safety of the ultrasound-guided intra-articular injection when performed according to the standardized technical procedure [20].

### Limits of the Study

A major limitation of our study is the heterogeneity and the different sample sizes of the two injection treatment groups, but this aspect can be justified by the design of the study, which was born as a real-world observational experiment.

A second important limitation in the interpretation of the results is the difficulty in determining in what qualitative and quantitative measure the association of hyaluronic acid with cortisone compared to hyaluronic acid alone is more or less effective in improving the clinical and functional outcomes.

## 5. Conclusions

Intra-articular hip injections have been considered in recent years a cornerstone of the conservative management of symptomatic hip osteoarthritis because they provide significant advantages by inducing a pain decrease, increasing the range of motion, improving functional ability, and reducing the consumption of analgesics. However, further studies are necessarily required to clarify fundamental aspects, such as the HOA-targeted infiltration protocol and the expected therapeutic effect in symptomatic hip OA.

The effects of CS in hip OA have been demonstrated in several controlled studies [46], although evidence is quite scarce, and many controversies remain because of its short-term effect and reports of adverse effects after injection [46,47] and deleterious effects on cartilage [48,49]. Potential toxicity of steroidson cartilage has been documented in knee OA [50], but there are no solid data regarding the potential effect on the hip cartilage, so it is not known if the supposed deleterious effect could be due to the steroid itself or the repeated injections.Therefore, further RCTs are needed.

Viscosupplementation might improve cartilage biochemistry with chondroprotective effects, even if these aspects have been shown only in vitro and not in human trials [51,52,53].

Regarding the injective schedule, it will be important to determine the ideal dose of CS, the most biologically suitable hyaluronic acid for association with steroids, but also the recommended rate and maximal number of allowed injections.

Regarding the therapeutic effect, targeted trials should also be conducted to establish a precise correlation between the type of injection and the type of hip osteoarthritis. The type of HOA means the radiological grade according to the Kellegren–Lawrence classification and the clinical phenotype based on the prevalence of one of the two main presenting elements of HOA: pain and functional impairment.

The growing availability of musculoskeletal US should allow a better and more precise evaluation of the hip environment to create a correlation between ultrasound prognostic elements (for example the degree of synovitis or cartilage thickness) and the corresponding infiltrative protocol.

The intrarticular administration of the association of hyaluronic acid and corticosteroids into the hip joint performed under ultrasound guidance could be a useful and safe instrument to reduce pain and improve function in symptomatic HOA in the longterm.

However, there is a need for more rigorously designed studies with a larger sample size to further define the evidence-based best practice for intra-articular treatment with the association of hyaluronic acid and corticosteroids for patients with hip osteoarthrosis, which is a subject that is still very much under debate.

## Figures and Tables

**Table 1 jcm-12-06600-t001:** Clinical data of 167 patients at baseline (T0).

Sex	
Male	41.90%
Female	58.10%
Mean age (years) (mean ± SD)	70.6 ± 12.2
Side of hip intra-articular injection	
Right	42.7%
Left	44.5%
Bilateral	12.8%
HA	
1500–2000 kDa	96%
620–1200 kDa	4%
HA + CS	76.20%
HA	23.80%
Kellgren–Lawrence	
1	0.70%
2	18.40%
3	63.90%
4	17.00%
VAS T0: (mean ± SD)	6.30 ± 2.36
GLFS T0: (mean ± SD)	60 ± 13.11

CS: corticosteroids; GLFS: Geriatric Locomotive Function Scale; HA: hyaluronic acid; VAS: Visual Analogue Scale.

**Table 2 jcm-12-06600-t002:** Questionnaire data at baseline (T0) and after 12 months (T1).

	T0	T1
	n. 167	n. 88
Drugs		
No drugs	42.50%	69.80%
Paracetamol	19.40%	10.50%
NSAIDs	33.10%	18.60%
Corticosteroids	5%	1.20%
Nutraceuticals		
No	73.30%	37.60%
Yes	26.70%	62.40%
Rehabilitation		
No	87.30%	37.50%
Yes	12.70%	62.50%
Physical Therapy		
No	96%	87.70%
Yes	4.00%	12.30%
Hip Arthroplasty		
No	99%	77.40%
Yes	0%	20.20%
Waiting list	1%	2.40%

**Table 3 jcm-12-06600-t003:** Outcome measures.

	T0 n. 167	T1 n. 88	*p* Level
VAS (mean ± SD)	6.30 ± 2.36	4.81 ± 2.58	*p* < 0.001
GLFS (mean ± SD)	60 ± 13.11	51.38 ± 13.11	*p* < 0.001

GLFS: Geriatric Locomotive Function Scale; VAS: Visual Analogue Scale.

**Table 4 jcm-12-06600-t004:** Analysis between groups.

		T0 (Mean ± SD) (HA + ST) n.127 vs. HA n.40	T1 (Mean ± SD) (HA + ST) n.79 vs. HA n.9
VAS	HA+ CS	6.54 ± 2.03 *	4.96 ± 2.9 ^
	HA	4.93 ± 3.17 *	4.81 ± 2.58 ^
GLFS	HA + CS	60.60 ± 12.87	52.50 ± 10.98
	HA	56.67 ± 13.82	48.61 ± 10.20

* *t* test between groups significant *p* level 0.001. ^ *t* test between groups significant *p* level 0.016.CS: corticosteroids; GLFS: Geriatric Locomotive Function Scale; HA: hyaluronic acid; VAS: Visual Analogue Scale.

## Data Availability

The data will be available on the SPSS database if requested.

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
