# Peer review of "The Effects of Ultrasound-Guided Intra-Articular Injections with Hyaluronic Acid and Corticosteroids in Patients with Hip Osteoarthritis: A Long-Term Real-World Analysis"

_jcm, 2023, doi:10.3390/jcm12206600_

Round 1
Reviewer 1 Report
1. There are indeed some spelling and formatting issues in this article, such as "ossess" in line 21, "arthroprotesis" instead of in Table 2, and missing punctuation in line 21. There are also inconsistencies in the abbreviations used, such as "CS" in line 182 and "IA" in line 204. The necessary changes are needed to improve the quality of the article. Additionally, the expression "p level" is not common in papers.
2. There are some issues with the tables that need to be addressed. Removing the first row of Table 1 or adding it to the table description should be considered. In the column of HA+CS in the statistics of Table 1, it is not necessary to label the value of “n” again. Considering to add a column to indicate injection dosage or concentration instead of using "YSE or NO," which is abrupt and inconsistent with the column above. Table 3 is not aligned correctly. Instead of directly using "VAS HA+ CS/ VAS HA," another column can be added to the left side of Table 4. Some annotations in the tables have first-line indentation, while others do not.
3. There are too many paragraphs in the introduction and discussion sections, which should be appropriately downsized or consolidated.
4. What does "waiting list" mean in Table 2? No concrete conclusions yet?
5. Were the authors able to specifically describe the approximate time and form of intervention received by the patients in the data collected at T0 and T1, long-term or short-term? To provide more details of the intervention content.
6. Under the conditions of large heterogeneity and different sizes of the sample, do the authors think that this study is of great significance? In addition, would this study be a better fit for meta-analysis by collecting more detailed data on other interventions received by the patients in this study?
Author Response
Dear Reviewer, We thank you for the suggestions. Below you will find explanations to your questions. 1. There are indeed some spelling and formatting issues in this article, such as "ossess" in line 21, "arthroprotesis" instead of in Table 2, and missing punctuation in line 21. There are also inconsistencies in the abbreviations used, such as "CS" in line 182 and "IA" in line 204. The necessary changes are needed to improve the quality of the article. Additionally, the expression "p level" is not common in papers. We corrected “ossess” with “assess” and "arthroprotesis" with arthroplasty. We checked the consistency of the abbreviations, as recommended, P level is useful to describe significant scores, as suggested in statistical analysis 2. There are some issues with the tables that need to be addressed. Removing the first row of Table 1 or adding it to the table description should be considered. In the column of HA+CS in the statistics of Table 1, it is not necessary to label the value of “n” again. Considering to add a column to indicate injection dosage or concentration instead of using "YSE or NO," which is abrupt and inconsistent with the column above. Table 3 is not aligned correctly. Instead of directly using "VAS HA+ CS/ VAS HA," another column can be added to the left side of Table 4. Some annotations in the tables have first-line indentation, while others do not. We modified the Tables according to your suggestions. 2. There are too many paragraphs in the introduction and discussion sections, which should be appropriately downsized or consolidated. We appropriately modified the introduction section 4. What does "waiting list" mean in Table 2? No concrete conclusions yet? Waiting list means the percentage of patients waiting for surgery 5. Were the authors able to specifically describe the approximate time and form of intervention received by the patients in the data collected at T0 and T1, long-term or short-term? To provide more details of the intervention content. Patients were evaluated at baseline before the infiltration treatment (T0), and after 12 months (T1) by telephone interview. Personal and clinical information (age, gender OA diagnosis date, traumatic events, comorbidities, current therapies), pharmacological therapies in progress, ongoing anti-inflammatory and pain-relieving therapies, rehabilita-tive treatments in progress related to hip osteoarthritis were collected. 6. Under the conditions of large heterogeneity and different sizes of the sample, do the authors think that this study is of great significance? In addition, would this study be a better fit for meta-analysis by collecting more detailed data on other interventions received by the patients in this study? Despite the heterogeneity and the different sample sizes of the two intervention groups, according to our practical experience it’s useful to assess the real efficacy and tolerability of IA injections in hip OA with the association of CS and HA, because this combination is usually used in clinical practice not only by our group, but also by other colleagues. The goal of this preliminary study is to describe the differences between hip injections with hyaluronic acid with or without corticosteroids.The next study step will be the collection of more detailed data for meta-analysis

Reviewer 2 Report
The brand and preparation method of hyaluronic acid should be determined.
What is the advantage of intra-articular ultrasound-guided injection compared to other intra-articular injection methods?
In line 151, what does mean the word steroids ?
In table 4, why is different n in T0 and T1 ?
a little editing of English language required
Author Response
Dear Reviewer, We thank you for the suggestions. Below you will find explanations to your questions. • The brand and preparation method of hyaluronic acid should be determined: we preferred not to insert brands, but rather to characterize the hyaluronic acids used based on their molecular weight, as shown in Table 1, as it is a no-funding study and there are many brands on the market and used over time • What is the advantage of intra-articular ultrasound-guided injection compared to other intra-articular injection methods? The hip is a deep joint and it is adjacent to important neurovascular structures, and non–image-guided injection might carry a risk of injury to the femoral artery, femoral nerve, and lateral femoral cutaneous nerve. For this reason ultrasonographic guidance is precious and useful in intra-articular hip injection, and in addition it allow a proper visualization of the needle during injection. The advantage of US guidance compared to fluoroscopic guidance is the absence of ionizing radiations and the visualization of soft tissue. No contrast is required for US-guided injections. • In line 151, what does mean the word steroids ? We used the term steroids as a synonimous for corticosteroids, but if it is confusing when reading the article, we replace it with “corticosteroids” • In table 4, why is different n in T0 and T1 ? Because of the initial 167 patients, evaluated at T0 (baseline), only 88 patients responded to the telephone interview 12 months after infiltration (T1).
